# A Reference Interval for CT-Based Liver Volume in Dogs without Hepatic Disease

**DOI:** 10.3390/vetsci11090400

**Published:** 2024-09-01

**Authors:** Reo Nishi, George Moore, Masahiro Murakami

**Affiliations:** 1Department of Veterinary Clinical Sciences, College of Veterinary Medicine, Purdue University, West Lafayette, IN 47907, USA; 2Department of Veterinary Administration, College of Veterinary Medicine, Purdue University, West Lafayette, IN 47907, USA

**Keywords:** liver size, hepatic volume, canine liver disease, liver segmentation

## Abstract

**Simple Summary:**

The measurement of canine liver size is essential, particularly in the evaluation of hepatic disease. Computed tomography (CT)-based liver volumetry can be useful for the assessment of liver size, but the reference interval has not been reported in dogs without hepatic disease. The purpose of the present study was to define the reference interval for CT-based liver volume normalized by body weight in dogs with no history of hepatic disease. The weight-based reference interval lower limit of 11.1–15.5 (90% confidence interval [CI]) to an upper limit of 31.9–42.6 (90% CI) cm^3^/kg was defined by evaluating CT scans of 121 dogs with no history of hepatic disease. This weight-based reference interval provides an accurate assessment of liver volume changes in dogs with various hepatic diseases, thereby facilitating the diagnosis and management of hepatic disease in veterinary medicine.

**Abstract:**

In both human and veterinary medicine, computed tomography (CT) volumetry provides a quantitative and accurate measure of liver volume. While CT volumetry is recognized as a useful method for assessing liver volume in dogs, a statistically significant reference interval for liver volume in dogs with no history of hepatic disease has not been reported. The purpose of the present study was to define a reference interval for liver volume with no history of hepatic disease using CT volumetry. Medical records from 2 June 2020 to 25 July 2022 were retrospectively reviewed, including 121 dogs that underwent abdominal CT scans and had no history of hepatic disease. Liver volumes were measured using CT volumetry and normalized by body weight. The median of normalized CT-based liver volume in 121 dogs was 22.2 cm^3^/kg. Based on these data, a weight-based reference interval lower limit of 11.1–15.5 (90% confidence interval [CI]) to an upper limit of 31.9–42.6 (90% CI) cm^3^/kg for CT-based liver volume was defined in dogs without hepatic disease. This study provides an accurate assessment of liver volume changes in dogs with various hepatic diseases.

## 1. Introduction

Liver size is an important diagnostic criterion in the evaluation of hepatic disease in veterinary medicine as well as in human medicine [1,2]. Radiography is commonly used to assess liver size in cases of suspected hepatic disease in dogs [1,3]. Despite its widespread use, the accuracy of radiography in detecting hepatomegaly or microhepatia in dogs is limited [4,5]. Therefore, there is a need for a more accurate method of measuring liver size, coupled with a reliable reference interval.

Computed tomography (CT) provides an accurate method for determining organ volumes, including in dogs [6]. CT-based liver volume estimates correlate highly with actual human liver weight, regardless of chronic hepatic disease etiology [7]. Thus, this non-invasive technique is increasingly recognized for its accuracy and utility in assessing hepatic function in both veterinary medicine and human medicine [6,7,8,9,10]. Evaluation of hepatic function is required before the following major surgeries: hepatectomy, portal vein embolization, and transplantation [11,12]. The correlation between CT-based liver volume and standard liver volume, as well as its association with liver function, has been established [12], suggesting the potential utility of CT-based liver volume in predicting hepatic diseases.

The application of hepatic volumetry derived by CT in dogs has been documented, notably in dogs with extrahepatic portosystemic shunt, where an increase in CT-based liver volume has been observed postoperatively [10,13,14]. Previous studies have reported the mean and range of liver volumes which were derived by CT and were normalized to body weight in dogs without hepatic disease [8,9]. However, these studies were limited by small sample sizes, which prevented the definition of a reliable reference interval for standardized CT-based liver volumes. In veterinary medicine, it is recommended to include over 120 cases to define a robust reference interval [15]. The definition of such a reference interval would increase the utility of CT volumetry and allow for its use in future studies to assess liver volume changes in various disease processes. Therefore, the purpose of this study is to define a reliable reference interval (95% confidence interval; CI) for CT-based liver volumes in a large sample of dogs with no history of hepatic disease.

## 2. Materials and Methods

### 2.1. The Case Selection

Dogs examined at the Purdue University Veterinary Teaching Hospital were retrospectively included in this study. Dogs that underwent abdominal CT between 2 June 2020 and 25 July 2022 were identified by searching the medical records database. Dogs with a history or suspected history of hepatic disease based on complete blood count, serum biochemistry (including alkaline phosphatase, alanine aminotransferase, total bilirubin, gamma-glutamyl transferase, glucose, and albumin levels), and clinical signs or history suggestive of liver disease were excluded. Clinical information such as breed, body weight, body condition score (BCS), and age was also recorded.

### 2.2. Case Evaluations Using CT Scans

CT scans were conducted using a 64-slice, third-generation CT scanner (Light Speed VCT, GE Medical Systems Inc., Waukesha, WI, USA). Comprehensive CT data, including the entire liver, were collected in dogs under general sedation or anesthesia. Scanning parameters included helical mode, 120 kVp, 280 mA, a pitch of 1, and slice thickness in the range of 1.25–3.75 mm, utilizing a precise algorithm. Post-contrast images were obtained at various intervals after the administration of a non-ionic iodinated contrast agent (Iohexol, OmnipaqueTM 240, GE Healthcare, Marlborough, MA, USA; 2.0 mL/kg), either in a single phase at 60 s or in three phases at 20, 55, and 95 s post-initiation of intravenous infusion. Both pre- and post-contrast liver images were evaluated by an ACVR board-certified radiologist (MM), and any dogs showing liver abnormalities, such as masses or nodules that could impact liver volume measurements on the CT images, were excluded. The remaining CT studies from dogs were considered as cases without hepatic disease. Only the pre-contrast images were used for subsequent CT liver volumetry.

### 2.3. Hepatic Volumetry Using Computed Tomography

A volumetry of the livers was conducted using CT by a veterinarian (RN) trained under the supervision of the radiologist (MM), using a DICOM viewer (Horos, version 4.0.0., https://horosproject.org [accessed on 12 January 2020]) in accordance with methodologies established in previous studies [8,9]. For all dogs, the window width and window level were adjusted to 350–400 HU and 40 HU, respectively. For liver segmentation, the entire liver was manually delineated into an operator-defined region of interest (ROI) in the transverse plane of the pre-contrast image, extending from the cranial border at the diaphragm to the caudal borders adjacent to the right kidney and spleen. Hepatic vessels within the hepatic parenchyma were included in the ROIs, while visible hepatic lobe fissures, the gallbladder, and hepatic vessels outside the hepatic parenchymal boundary were excluded according to previous reports [8,9]. To enhance efficiency, liver margins were traced instead of filling in the liver parenchyma (Figure 1). After manually outlining the ROIs on more than 20 slices with equal interslice gaps, the liver volume derived by CT was obtained using the following formula: Σ {each slice area (cm^2^) × slice thickness (cm)} × total number of slices of hepatic parenchyma/number of slices used for calculation, following the protocols from previous reports [8]. Finally, the CT-based liver volume was divided by the dog’s body weight (kg) to determine the normalized CT-based liver volume for each case, as body weight has been reported to be the best normalization factor for liver volumetry [9].

### 2.4. Statistics

The analyses were conducted using open-source statistical software (EZR version 1.61, Jichi Medical University, Saitama, Japan; graphical user interface for R, The R Foundation for Statistical Computing, Vienna, Austria) and MedCalc statistical software version 12.7.5.0 (MedCalc Software Ltd., Ostend, Belgium). Statistical significance was determined at a *p*-value threshold of 0.05.

The Shapiro–Wilk normality test was first performed on the liver volume derived by CT and body weight in all cases, revealing a non-parametric distribution. Consequently, the correlation between CT-based liver volume and body weight was analyzed using Spearman’s rank correlation, which is appropriate for non-parametric variables.

Following reference interval guidelines [15], the robust method was used to provide a 90% confidence interval (CI) for the limits of the normalized CT-based liver volume using MedCalc statistical software. The Shapiro–Wilk normality test was repeated for the normalized CT-based liver volume, confirming a non-parametric distribution. Subsequently, log transformation of the data was performed, and the Shapiro–Wilk test confirmed the log-transformed data to be parametric. The log-transformed means ± 1.96 standard deviations (SDs) were then calculated and reconverted to determine the 95% CI for the normalized CT-based liver volume.

To assess the effect of BCS on normalized CT-based liver volume, dogs were classified into three BCS categories: low (BCS 1 to 3), normal (BCS 4 to 6), and high (BCS 7 to 9). For each group, the number of dogs was recorded, and the median and range of normalized CT-based liver volumes were calculated.

## 3. Results

This study included a total of 121 dogs, and the laboratory test results and clinical histories were assessed. The dogs ranged in body weight from 2.2 kg to 104.3 kg (median, 38.0 kg; Table 1) and in age from 0.6 to 15.0 years (median, 7.0 years) at the CT scan. The breeds included 24 mixed-breed dogs, 16 German Shepherd dogs, 11 Golden Retrievers, 7 Labrador Retrievers, 6 Goldendoodles, 4 Great Danes, 4 Rottweilers, 3 Belgian Malinois, 3 Old English Mastiffs, 3 Siberian Huskies, 2 Chesapeake Bay Retrievers, 2 Bull Mastiffs, 2 Dachshunds, 2 Doberman Pinschers, 2 English Bulldogs, 2 French Bulldogs, 2 Mastiffs, 2 Saint Bernards, and 1 each of the following breeds: Akita, Alaskan Malamute, American Cocker Spaniel, Australian Shepherd, Belgian Sheepdog, German Shorthaired Pointer, Brussels Griffon, Cane Corso, Cardigan Welsh Corgi, Chihuahua, Cockapoo, Colie, Coonhound, English Shepherd, French Mastiff, Giant Schnauzer, Great Pyrenees, Havanese Terrier, Jack Russel Terrier, Labradoodle, Plott Hound, Redbone Coonhound, Shih Tzu, and Weimaraner.

The median CT-based liver volume was calculated as 774.8 cm^3^ in dogs without hepatic disease (Table 1). There was a strong correlation (correlation coefficient, 0.81) between CT-based liver volume and body weight (*p* < 0.001, Figure 2) using Spearman’s rank correlation coefficient.

The median CT-based liver volume normalized by body weight was 22.2 cm^3^/kg, with a range of 11.1 to 42.7 cm^3^/kg (Table 1). Using the non-parametric percentile method, the 90% CIs for the lower and upper reference limits of the normalized CT-based liver volume were determined to be 11.1–15.5 cm^3^/kg and 31.9–42.6 cm^3^/kg, respectively.

After reconverting the log-transformed data, the 95% CI for the weight-based liver volume using CT in dogs with no history of hepatic disease in this study was calculated to be 14.3–34.7 cm^3^/kg (Table 2). Therefore, the weight-based reference interval for normalized CT-based liver volume was defined as 11.1–15.5 (lower limit 90% CI) to 31.9–42.6 (upper limit 90% CI) cm^3^/kg in dogs without hepatic disease. Notably, the lower and upper limits of this interval fell within the 90% CIs for the reference limits (Table 2).

Out of 121 dogs, the body condition score (BCS) was not reported for 2 dogs, leaving 119 dogs with documented BCSs. Of these, 98 dogs (82.4%) had a normal BCS (BCS of 4 to 6), 18 dogs (15.1%) had a high BCS (BCS of 7 or 8, with no dogs scoring a BCS of 9), and 3 dogs (2.5%) had a low BCS (BCS of 3, with no dogs scoring 1 or 2). The median normalized CT-based liver volume in the normal BCS group was 22.5 cm^3^/kg (range: 11.1 to 42.7 cm^3^/kg), which was similar to the median for all dogs in this study (22.2 cm^3^/kg, range: 11.1 to 42.7 cm^3^/kg). The median normalized CT-based liver volume in the low BCS group was 28.2 cm^3^/kg (range: 23.0 to 39.5 cm^3^/kg), while in the high BCS group, it was 21.2 cm^3^/kg (range: 15.5 to 25.4 cm^3^/kg).

## 4. Discussion

This study successfully defined a weight-based reference interval for CT-based liver volume in dogs, based on data from 121 individuals. The median CT-based liver volume was 774.8 cm^3^, and the median normalized CT-based volume was 22.2 cm^3^/kg, with a weight-based reference interval of 11.1–15.5 (lower limit 90% CI) to 31.9–42.6 (upper limit 90% CI) cm^3^/kg.

This study included dogs of various sizes and breeds without biochemical or clinical suspicion of liver disease, with body weights ranging from 2.2 to 104.3 kg. This range provides a wider variability compared to a previous study (6.5 to 84.2 kg) [9] and is considered to be more representative of the general population. Body weight is known to be a useful normalizing factor for liver volumes using CT volumetry in dogs [9,10,13,14,15]. In our study, a strong positive correlation was shown between CT-based liver volumes and body weight, consistent with previous studies [9,10]. Therefore, the weight-based reference interval was defined using CT-based liver volumetry.

The definitive values of the 95% CIs for weight-based liver volume were determined in 121 dogs without liver disease using the robust method in the current study. Although the sample size of more than 120 dogs was sufficient to define the range of the lower and upper limits of the reference interval, larger sample sizes are recommended to establish more definitive values for the reference interval [15]. In this study, we reported the 90% CIs for the upper and lower limits, specifically 11.1–15.5 cm^3^/kg (lower limit, 90% CI) to 31.9–42.6 cm^3^/kg (upper limit, 90% CI). While the definitive 95% CI for the normalized CT-based liver volume of 14.3–34.7 cm^3^/kg fell within the calculated 90% CIs of the reference limits, it is important to note that these values may change within the 90% CIs with larger sample sizes.

The results of the present study are consistent with previous studies that used smaller sample sizes. For instance, Stieger et al. reported a mean (±SD) CT-based liver volume normalized by body weight of 24.5 (±5.6) cm^3^/kg in six dogs without hepatic disease [10]. Similarly, Kinoshita et al. conducted CT volumetry in 41 dogs with no history of hepatic disease, reporting a mean (±SD) liver volume normalized by body weight of 22.1 (±4.6) cm^3^/kg [9]. However, the limited sample sizes in these studies precluded the definition of a reliable reference interval for liver volume based on CT-based data, given the recommendation in veterinary medicine to include at least 120 cases to define a reference interval [15]. Our study measured liver volumes in 121 dogs and the median (range) normalized CT-based liver volume was found to be 22.2 (11.1–42.7) cm^3^/kg. Although the method of tracing liver margins to measure liver volume differed slightly from that used by Kinoshita et al. [9], our results are consistent with previous reports. The larger sample size in the present study enabled the definition of a reliable reference interval for the weight-based liver volume in dogs without hepatic disease using CT volumetry, with a 95% CI of 14.3 (90% CI: 11.1–15.5) to 34.7 (90% CI: 31.9–42.6) cm^3^/kg.

In a previous report [9], it was suggested that normalized liver volume may be influenced by BCS, and accounting for BCSs in evaluations could help define a more accurate and narrower normal reference range. To investigate this, we incorporated BCS data into our study, categorizing the dogs into three groups: low, normal, and high BCSs. The majority of the dogs (82.4%) were in the normal BCS group, and the median normalized CT-based liver volume for this group was nearly identical to the value derived from the entire population. Interestingly, the smallest and largest normalized liver volumes were both observed within the normal BCS group, resulting in the same reference range for this group and the overall population. The median normalized liver volumes for the low and high BCS groups were larger and smaller, respectively. While the small sample sizes in these groups limit the ability to draw definitive conclusions, our findings align with previous research [9], where a higher BCS was associated with lower normalized liver volumes, potentially leading to inaccuracies in dogs with high BCSs. The small number of dogs in the low and high BCS groups and the similarity in the median and range of normalized liver volumes between the normal BCS group and the overall population support the validity of our results.

One limitation of the present study is the lack of a necropsy or liver biopsy to definitively exclude hepatic disease. However, given the limitation, this methodology represents the optimal approach achievable and should be considered acceptable. Future studies should aim to include histopathologic confirmation to further validate the reference interval.

Another limitation of our study is the relatively wide 95% CI for normalized CT-derived liver volume, which is comparable to or wider than previous reports with smaller sample sizes. Although a larger sample size generally improves precision and narrows the CI, our results showed a CI that overlapped with, and occasionally exceeded, those from prior studies. This may be due to our inclusion of a more diverse population, potentially encompassing a broader range of body sizes and conformations. This increased variability may have contributed to the wider CI. To address this, future studies focusing on specific breeds or body conformations may help reduce the CI for normalized canine liver volume, improving the precision and clinical applicability of these measurements.

## 5. Conclusions

This study defined a reliable CT-based reference interval for weight-based liver volume in dogs without hepatic disease, with a median of 22.2 cm^3^/kg and a 95% confidence interval of 14.3 (90% CI: 11.1–15.5) to 34.7 (90% CI: 31.9–42.6) cm^3^/kg. This weight-based reference interval provides an accurate assessment of liver volume changes in dogs with various hepatic diseases, thereby facilitating the diagnosis and management of hepatic disease in veterinary medicine.

## Figures and Tables

**Figure 1 vetsci-11-00400-f001:**
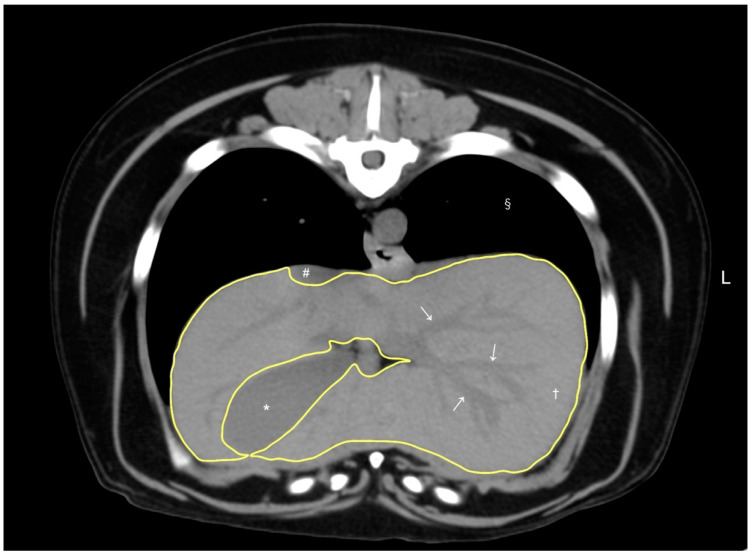
A representative image of abdominal CT transverse pre-contrast images using CT liver volumetry in dogs. The window width and window level were adjusted to 350–400 HU and 40 HU, respectively. The liver segmentation was manually traced as a region of interest (ROI, yellow line). Hepatic vessels (white arrows) within the hepatic parenchyma (†) were included in the ROI, whereas the gallbladder (*), visible liver lobe fissures, and hepatic vessels outside the hepatic parenchymal margin were excluded. The caudal vena cava (#) and pulmonary parenchyma (§) were also noted.

**Figure 2 vetsci-11-00400-f002:**
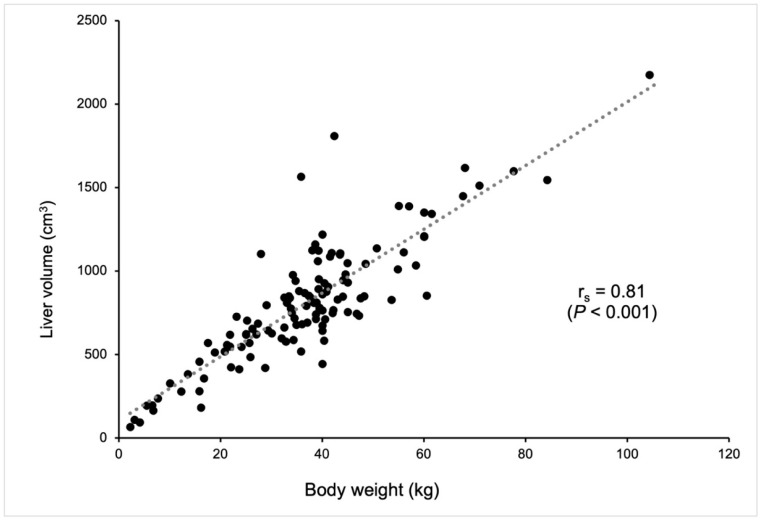
The correlation between CT-based liver volume (cm^3^) and body weight (kg). Each dot represents an individual dog. A strong positive correlation (Spearman’s rank correlation coefficient, r_s_ = 0.81) was observed between liver volume and body weight, indicating that larger dogs tend to have larger liver volumes. The correlation was statistically significant (*p* < 0.001). The dotted line represents the trend line of the correlation.

**Table 1 vetsci-11-00400-t001:** Descriptive statistics for body weight and CT-based liver volume in 121 dogs without hepatic disease.

	Median (Range)
Body weight (kg)	38.0 (2.2–104.3)
Liver volume (cm^3^)	774.8 (66.2–2174.8)
Liver volume/body weight (cm^3^/kg)	22.2 (11.1–42.7)

**Table 2 vetsci-11-00400-t002:** The weight-based reference interval of reconverted values of log-transformed normalized liver volume in dogs without hepatic disease.

	95% CI in 121 Dogs (90% CI of Lower and Upper Limit)
Liver volume/body weight (cm^3^/kg)	14.3 (11.1–15.5)–34.7 (31.9–42.6)

## Data Availability

The data that support the findings of this study are available from the corresponding author upon reasonable request.

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
