# Peer review of "A Reference Interval for CT-Based Liver Volume in Dogs without Hepatic Disease"

_vetsci, 2024, doi:10.3390/vetsci11090400_

Round 1
Reviewer 1 Report
Comments and Suggestions for Authors
In this submitted manuscript, the authors highlight the opportunity of calculating the liver volume in healthy dogs based on CT images. However, while the authors propose to use accurate measurements the analyses techniques do leave some room for interpretation in that regard which could increase the impact of this manuscript substantially.
I have some questions regarding the manuscript:
- line 79-80: The CTs are obtained with a certain slice thickness, why do you reconstruct it with a thicker slice (because now you might loose some interesting details) and what is the pitch used to obtain the images?
- line 81: why do you describe the post-contrast images when you don't use them?
- line 85: what kind of liver abnormalities were detected and in how many dogs? Because I don't expect that all abnormalities have an impact on the volume as they are just 'normal'. In addition, how do you define a liver abnormality?
- line 95: why did you use a manual deleating method and did you check the intra-observer variation?
- line 102: do you mean that you delineated at least 20 slices or that you exactly delineated 20 slices? Are these than next to each other? This means that you cover a length of 10 cm (when a pitch of 1 is applied). I can imagine that this too long for a dog of 2.2 kg. How did you compensate for this?
- line 103: you obtained a 3D image but now you reconstruct it in a 2D manner and than calculate it back to a 3D volume. This is a pity there are programms available by which it is not necessary to use a formula anymore.
line 105: what is the additional value of the normalized CT-based liver volume?
line 177-180: fully agree that this is a broad range and that this is more representative though is than also more clinical relative? As the range is relatively large do you know what it means when someone is outside the range or on the boundaries of the range? Why don't you use a formula based on the correlation and say that with this weight this volume is normal with a range of 10-20%?
- line 204: for consistency add the interfall similar to line 197 and line 199.
- line 206: I would ellaborate a bit more on the fact that the reference interval is expected to become smaller with larger sample sizes though in this study it is larger in comparison with two other studies.
General remark regarding the results section, is there a reason why you decided not to dive a bit more into the results by splitting the group of animals based on type, weight and age. Not to create significant results but to see if there are trends/suggestions or differences?
Comments on the Quality of English Language
I don't have any major comments on the English language
Reviewer 2 Report
Comments and Suggestions for Authors
Brief summary
This study is based on the use of computed tomography to determine liver volume in dogs. The main strength of the study is the number of animals used in the study compared to previous studies.
The main weakness is that the Body Condition Score (BCS) of the study animals is not taken into consideration, based exclusively on their weight. This is a very important source of bias, particularly because the weight range presented is very high. You will agree with me that a 28 kg Cocker Spaniel is not the same as a 28 kg German Shepherd. The BCS is the tool that homogenizes body weight and should be used in studies as a mechanism for homogenizing the sample.
Inaccuracy
There is an inconsistency in the text. In the abstract and summary it refers to a 90% confidence interval, but then in the objectives and results it talks a 95% confidence interval. Which one is correct? Please check this.
There is no reference to the clients' authorization for the inclusion of the animals and the results in the study.
Specific comment
To provide consistency the results should incorporate the BCS of the animals studied and, if the objective is to determine the normality of liver size/volume, those animals with BCS considered not normal should be excluded; for example, exclude those with BCS > 6 and BCS < 4; or any other criteria you consider more correct to exclude dogs with obesity or too thin. These animals may have hepatic fatty infiltration, or the opposite, causing an error in the calculations.
Round 2
Reviewer 1 Report
Comments and Suggestions for Authors
I would like to thank the authors for their valuable comments and alterations in the manuscript.
Comments on the Quality of English Language
Reviewer 2 Report
Comments and Suggestions for Authors
Thank you for taking my suggestions for improvement into consideration. The possible error paths have been corrected to the best of my knowledge.